# Antifungal potential of *Rhinacanthus nasutus* extracts against the pathogenic fungus *Cryptococcus neoformans*

**Muthita Khongthongdam**[1,2☯]**, Trinity Jantarach**[1,2,3☯]**, Tanaporn Phetruen**[1,2¤]**, Sittinan Chanarat**[1,2*]

1 Laboratory of Medical Molecular Mycology, Department of Biochemistry, Faculty of Science, Mahidol University, Bangkok, Thailand, 2 Center for Excellence in Protein and Enzyme Technology, Faculty of Science, Mahidol University, Bangkok, Thailand, 3 Bangkok Patana School, Bangkok, Thailand

☯ These authors contributed equally to this work.
¤ Current Address: Department of Pharmacology, Faculty of Science, Mahidol University, Bangkok, Thailand
* sittinan.cha@mahidol.ac.th

## Abstract

*Rhinacanthus nasutus* (snake jasmine) has been widely used in Thai-traditional medicine to treat various ailments, including fungal infections. This study aimed to investigate the antifungal activity of *R. nasutus* extracts and their impact on the virulence factors of *Cryptococcus neoformans*, a pathogenic yeast responsible for cryptococcal infections in immunocompromised individuals. Extracts were prepared from the leaves, roots, and a combination thereof, with rhincanthin-C quantified using High-Performance Liquid Chromatography (HPLC). Antifungal activity was assessed using a spot assay against *Saccharomyces cerevisiae* and *C. neoformans*. The root extract demonstrated the strongest antifungal effect, particularly against *C. neoformans*. Further investigation into virulence factors revealed a noticeable reduction in melanin production, cell and capsule sizes and alterations in cell structure, suggesting cellular abnormality. The findings of this study demonstrate the antifungal potential of *R. nasutus* and may warrant future exploration of its therapeutic applications for treating fungal infections.

## Introduction

*Rhinacanthus nasutus,* commonly known as "snake jasmine", is a small shrub, 0.6–1.2 meters tall, native to tropical regions of Southeast Asia [1]. It has long been valued in traditional medicine, where different preparations are used to treat a variety of conditions. These include external skin conditions like ringworm, psoriasis, eczema, and internal conditions, such as hepatitis, diabetes, and hypertension [1–3]. Specific plant parts have been used for different purposes: for instance, root

**Data availability statement:** All relevant data are within the manuscript and its Supporting Information files.

**Funding:** SC received fundings from CIF and CNI Grant (Faculty of Science, Mahidol University), MU's Strategic Research Fund: fiscal year 2023 (MU-SRF-RS-07A/67), and Mahidol University (Fundamental Fund: fiscal year 2025 by National Science Research and Innovation Fund (NSRF: FF-107/2568). Funder's website: www.mahidol.ac.th The funders had no role in study design, data collection and analysis, decision to publish, or preparation of the manuscript.

**Competing interests:** No authors have competing interests.

decoctions are used as antidotes for snake bites, while the leaf extracts for heat rashes [1,4].

Prior studies have shown that extracts from *R. nasutus* possess antimicrobial, antioxidant, and anticancer properties [5]. Rhincanthin-rich leaf extracts have demonstrated antibacterial activity against *Staphylococcus epidermidis*, *Streptococcus mutans, Propionibacterium acnes,* and *Staphylococcus aureus* [6–8]. The methanolic extracts of *R. nasutus* have exhibited potent antioxidant activity in cell-free in vitro studies and have reduced oxidative stress in animal models [9]. *R. nasutus* has also shown anti-cancer properties in human oral cancer cells by inducing apoptosis and cell cycle arrest through modulation of Akt/p38 signaling pathways [10]. Furthermore, leaf extracts of *R. nasutus* have shown fungicidal activity by targeting the cell wall, leading the degeneration of the cell and death [1,2]. Specifically, *R. nasutus* has been reported inhibit the growth of various fungal species including *Microsporum gypseum*, *M. canis*, *Trichophyton rubrum*, *T. mentagrophytes*, *Epidermophyton floccosum*, *Candida albicans*, *Aspergillus niger*, *Malassezia sp.*, and *Magnaporthe oryzae* [8,11–13].

*Cryptococcus neoformans*, a globally distributed pathogenic yeast, is the first ranking species the critical priority category in the World Health Organization's Fungal Priority Pathogens List (FPPL) [14]. Upon being inhaled from the environment, *C. neoformans* cells can cause human infections, by initially affecting the lungs and progressing to the blood to cause cryptococcaemia and the central nervous system to cause cryptococcal meningitis [14–17]. The majority of patients with cryptococcal diseases are immunocompromised, with HIV infection being the primary risk factor. Organ transplant recipients and individuals taking immunosuppressive medications are also at risk, and infection can occur in seemingly healthy individuals. Cryptococcosis caused by *C. neoformans* is a severe disease, with mortality rates ranging from 41% to 61%, particularly in HIV positive patients, indicating the need for effective treatment options [18].

Current treatment options for cryptococcosis are limited. Localized cases can be treated with fluconazole, while severe and disseminated cases require treatment with amphotericin B in combination with flucytosine, followed by fluconazole [14,16,19,20]. Although these treatments are listed in the World Health Organization Essential Medicines List, but they remain inaccessible in many countries, and antifungal resistance is an emerging concern [21]. Indeed, reports of reduced fluconazole susceptibility have already been documented [22]. Moreover, amphotericin B—the frontline drug for severe cryptococcosis—is associated with significant nephrotoxicity, often limiting its safe use [23]. These limitations underscore the urgent need to identify new antifungal candidates, including plant-derived compounds with novel mechanisms of action.

Given these limitations, exploring novel antifungal agents is essential. In this study, we investigate the antifungal activity of *R. nasutus* extracts against *C. neoformans* and assess their impact on key virulence factors. These findings may help lay the groundwork for developing plant-based therapeutic options for cryptococcosis and related fungal infections.

## Materials and methods

### Preparation of *R. nasutus* extract

Leaves and roots of *R. nasutus* were collected between February to July 2024 from the Rammasak Women's Herbal Group Farm, located in Rammasak Subdistrict, Ang Thong Province, Thailand (14.6954°N, 100.2719°E). The plants were harvested at 12–14 months of age, within their mature growth stage. The harvesting process involved uprooting the entire plant and thoroughly washing the roots and leaves to remove all dirt. Taxonomic identification of the plant material as *R. nasutus* was confirmed by Prof. Dr. Paweena Traiperm, a specialist in plant taxonomy from the Department of Plant Science, Faculty of Science, Mahidol University.

Plant materials were oven-dried at 52–55°C for one week, ground into fine powder, and extracted following a modified protocol from previous studies [24,25]. Three types of extracts were prepared: leaf-only, root-only, and combined leaf and root. For single-part extracts, 5 g of powdered leaf or root were dissolved in 20 mL of solvent (25% glycerol in ethanol). For combined extracts, 2.5 g of each powder were mixed in the same solvent. The resulting extract concentration was calculated based on the dry weight of plant material relative to solvent volume, yielding 250 mg/mL (w/v). All mixtures were shaken at room temperature for 3 days, filtered through Whatman grade 5 paper (2.5 µm pore size), wrapped in foil, and stored refrigerated until use.

### HPLC analysis

The rhinacanthin-C content was quantified using High-Performance Liquid Chromatography (HPLC) under isocratic conditions [24,25]. The analysis was conducted on a Nova-Pak C18 60Å, 4 µm column. The mobile phase consisted of a mixture of methanol (Sigma-Aldrich; 34860) and 5% aqueous acetic acid (Sigma-Aldrich; V800018) in an 80:20 (v/v) ratio. A 1.0 mL/min flow rate was maintained, and the injection volume was 20 µL. Detection and quantification of rhinacanthin-C were carried out at a wavelength of 254 nm.

### Fungal strains and culture conditions

*Saccharomyces cerevisiae* strain W303, and *Cryptococcus neoformans* strain KN99α (S1 Table) were cultured in YPD (1% yeast extract (Himedium; RM027), 2% peptone (Himedium; RM001), and 2% glucose (Sigma-Aldrich)) medium [26–28]. The cultures were incubated at 30°C for 2–3 days for optimal growth and sufficient cell density for further experimentation.

### Antifungal testing using spot assay

To evaluate the antifungal activity of *R. nasutus* extract, we tested it against two fungal species: *S. cerevisiae*, and *C. neoformans.* A crude extract stock (250 mg/mL, calculated as plant dry weight per solvent volume) was diluted with YPD medium to final concentrations of 16, 8, and 4 mg/mL. Fungal cultures were adjusted to an initial $OD_{600}$ of 0.2, followed by 10-fold serial dilutions. For each dilution, 2.5 µL of the cell suspension was spotted onto YPD agar plates containing the respective extract concentrations. Plates were incubated at 30°C for 2 days, with growth recorded daily.

### Checkerboard assay

Synergy checkerboard assay was used to evaluate the interaction between *R. nasutus* extract and the antifungal drugs commonly used as first-line treatments for *C. neoformans*, including fluconazole (FLZ; Tokyo Chemical Industry; F0750), amphotericin B (AMB; Sigma-Aldrich; A9528), and 5-fluorocytosine (5FC; Sigma-Aldrich; F7129). The 96-well plate was prepared by diluting the two compounds in RPMI-1640 media (Invitrogen; 31800022), as described previously [19]. The final concentration in the 96-well plate ranged between 0.25–16 µg/mL for FLZ, 0.03125–2 µg/mL for AMB, 0.0078–0.5 µg/mL for 5FC, and 0.03125–1 mg/mL for *R. nasutus* root extract (expressed as plant dry weight per final volume). The

*C. neoformans* inoculum was prepared by diluting the culture to a 0.5 McFarland standard, and 10 µL of this suspension was added to each well of the 96-well plate. The plate was then incubated at 30°C for 2 days. The results were measured using a microplate reader at an absorbance of 600 nm.

The following equation quantified the degree of interactions, represented by the Fractional Inhibitory Concentration, or FIC index:

$$\frac{A}{MIC_A} + \frac{B}{MIC_B} = FIC_A + FIC_B = FIC\ Index$$

where the minimum inhibitory concentration (MIC) is defined as the lowest concentration of a test compound that shows no fungal growth, comparable to the blank RPMI-1640 medium. In this equation, A and B represent the MICs of each antibiotic in combination (in a single well), while $MIC_A$ and $MIC_B$ are the MIC of each drug individually. The FIC index value is then used to categorize the interaction of the two antibiotics tested: synergy (< 0.5), additive or indifference (0.5–4), and antagonism (> 4) [29,30].

### Virulence factors of *C. neoformans*

Melanin production was assessed in minimal medium (MM: 15 mM glucose (Sigma-Aldrich), 10 mM $MgSO_4$ (Glentham life sciences; 10034-99-8), 29.4 mM $KH_2PO_4$ (Phyto technology laboratories; 7758-11-4), 13 mM glycine (Lobachemie pvt.ltd.; 03976), and 3 mM thiamine (Sigma life science; T1270), MM + solvent, and MM + *R. nasutus* root extract. L-DOPA (Merck; D9628) was added where indicated to induce melanin [26,31]. Culture were incubated for 3 days with daily observation across six conditions. For quantification, spots were imaged under identical exposure settings, and melanin intensity was measured from digital micrographs using ImageJ (v1.51) software [32]. Pixel intensity values were recorded on an 8-bit grayscale scale (0 = black, 255 = white) and plotted as histograms to compare treated and control groups. Lower pixel intensity values indicated higher melanin pigmentation.

For cell morphology and capsule induction, cultures were grown in YPD medium (cell size) or 1/10 Sabouraud dextrose broth (SDB: 0.4% glucose (Himedia; RM027), and 0.1% peptone (Himedia; RM001)) with 5 mM MOPS pH 7.3 (capsule induction) [26,33,34]. Conditions included medium alone, medium + solvent, and medium + *R. nasutus* root extract. Cultures were incubated at 37°C with 5% $CO_2$, and observations were made on days 3 and 5.

Cells were stained with India ink for capsule visualization. Cell diameter (excluding capsule) and capsule thickness (distance from cell wall to capsule edge) were measured in ImageJ (v1.51) from 50 randomly selected cells per condition (*n* = 50). Mean values and distributions were reported for days 3 and 5 [26,33,34].

### Statistical analysis

All experiments were conducted in duplicate biological replicates (*n* = 2). Quantitative data (absorbance, cell diameter, capsule thickness) are presented as mean ± standard deviation (SD). Comparisons between treatment and control groups were analyzed using unpaired Student's *t*-tests in GraphPad Prism v10, with *p* < 0.05 considered statistically significant. All raw data supporting the analyses presented in this study can be found in the S1 File.

## Results

### Preparation and HPLC analysis of *Rhinacanthus nasutus* extracts

We prepared three types of extracts for analysis: leaf extract, root extract, and a combined leaf and root extract. After three days of shaking and sterile filtration, we obtained extracts with different appearances: leaf extract was dark green, the root extract dark red, and the combined extract dark brown (Fig 1A and B). Given that all three extracts have absorption at 254 nm, this wavelength was utilized for quantification. Various mixtures of methanol and 5% aqueous acetic acid

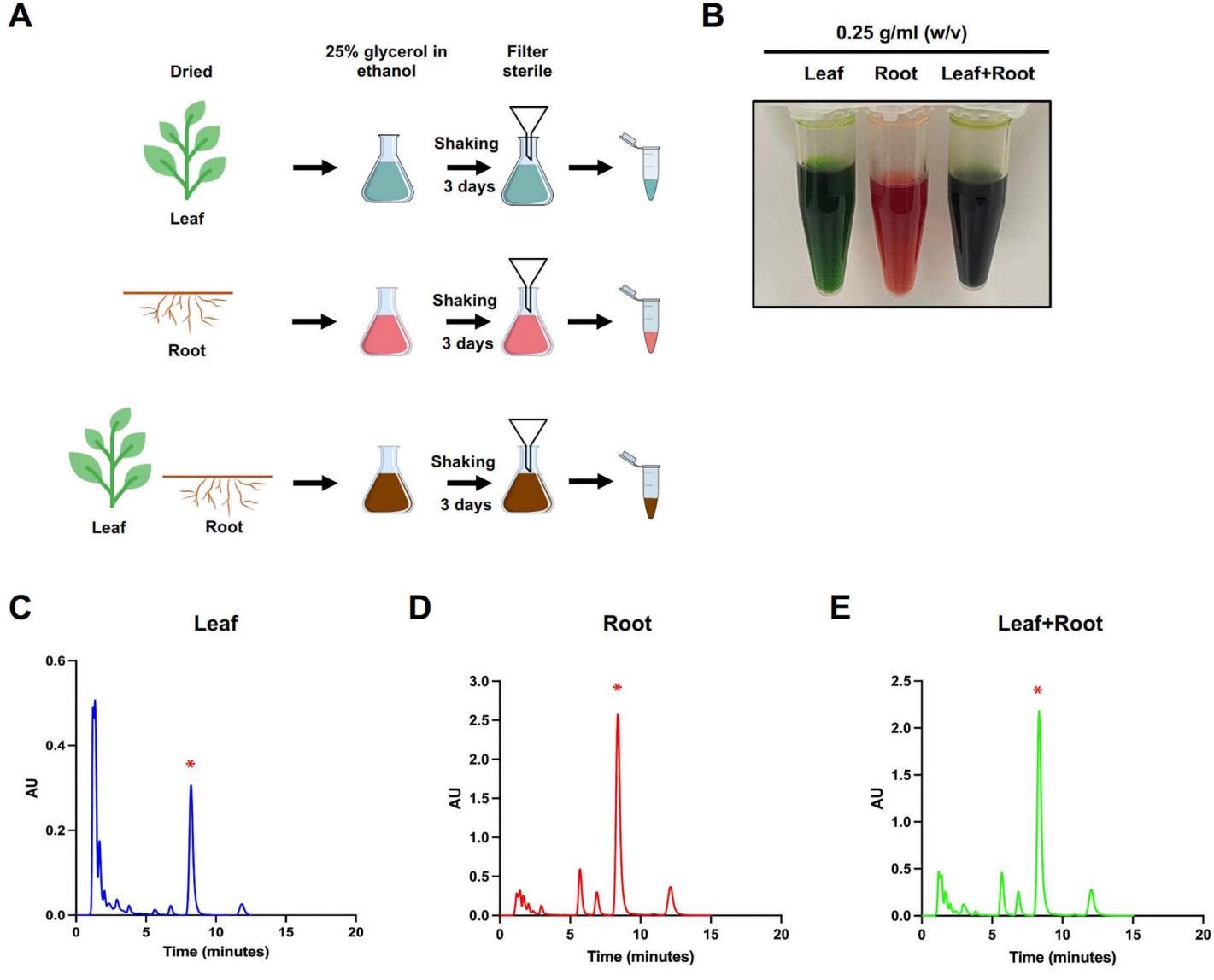

**Fig 1. Extraction process and HPLC analysis of *Rhinacanthus nasutus*. (A)** Schematic diagram illustrating the extraction process of *R. nasutus* using 25% glycerol in ethanol. Three sample types were prepared: leaf extract (5 g of dried leaves), root extract (5 g of dried roots), and a combined leaf-root extract (2.5 g of each). The samples were subjected to shaking for three days, followed by sterile filtration. **(B)** Visual appearance of extracts of *R. nasutus*. The leaf extract appeared dark green. The root extract appeared dark red. The combined extract appeared dark brown. **(C–E)** HPLC chromatograms of *R. nasutus* extracts: **(C)** leaf extract, **(D)** root extract, and **(E)** combined leaf-root extract, showing the unique chemical profiles of the respective extract types. Asterisks denote the putative peak corresponding to rhinacanthin-C.

were evaluated as the mobile phase, and the composition was optimized. It was determined that a ratio of 80:20 v/v of methanol to 5% aqueous acetic acid was necessary to achieve good resolution of the *R. nasutus*. All three extracts were eluted within 15 minutes.

Our chromatographic analysis showed that the root extract exhibited the strongest peak (2.6 AU), followed by the combined extract (2.2 AU) and the leaf extract (0.32 AU) (Figs 1C–E and S1 Fig). The elution pattern was consistent with previous reports using the same extraction protocol [24,25], supporting the identification of the dominant peak as the major

compound rhinacanthin-C. Although absolute quantification was not possible due to the absence of a commercially available standard and our inability to synthesize the compound, we estimated its concentration to be approximately 3.3 mg/mL in the leaf extract and 25 mg/mL in the root extract, based on reported values obtained using the same extraction strategy [24,25]. These results indicate that the root extract contains the highest level of rhinacanthin-C among the preparations.

### Antifungal activity of *R. nasutus* extract on yeast growth

The antifungal activity of the *R. nasutus* extracts was evaluated against two fungal species: *Saccharomyces cerevisiae* (Sc), and *Cryptococcus neoformans* (Cn). A spot assay was conducted on YPD medium containing leaf, root, and combined leaf-root extracts. We observed strong inhibition of fungal growth, particularly with the root extract, against both *S. cerevisiae* and *C. neoformans* (Fig 2A and S2 Fig). Given the pronounced potency of the root extract and its strong effects on *C. neoformans*, further experiments focused solely on the root extract and limited the study to *C. neoformans*.

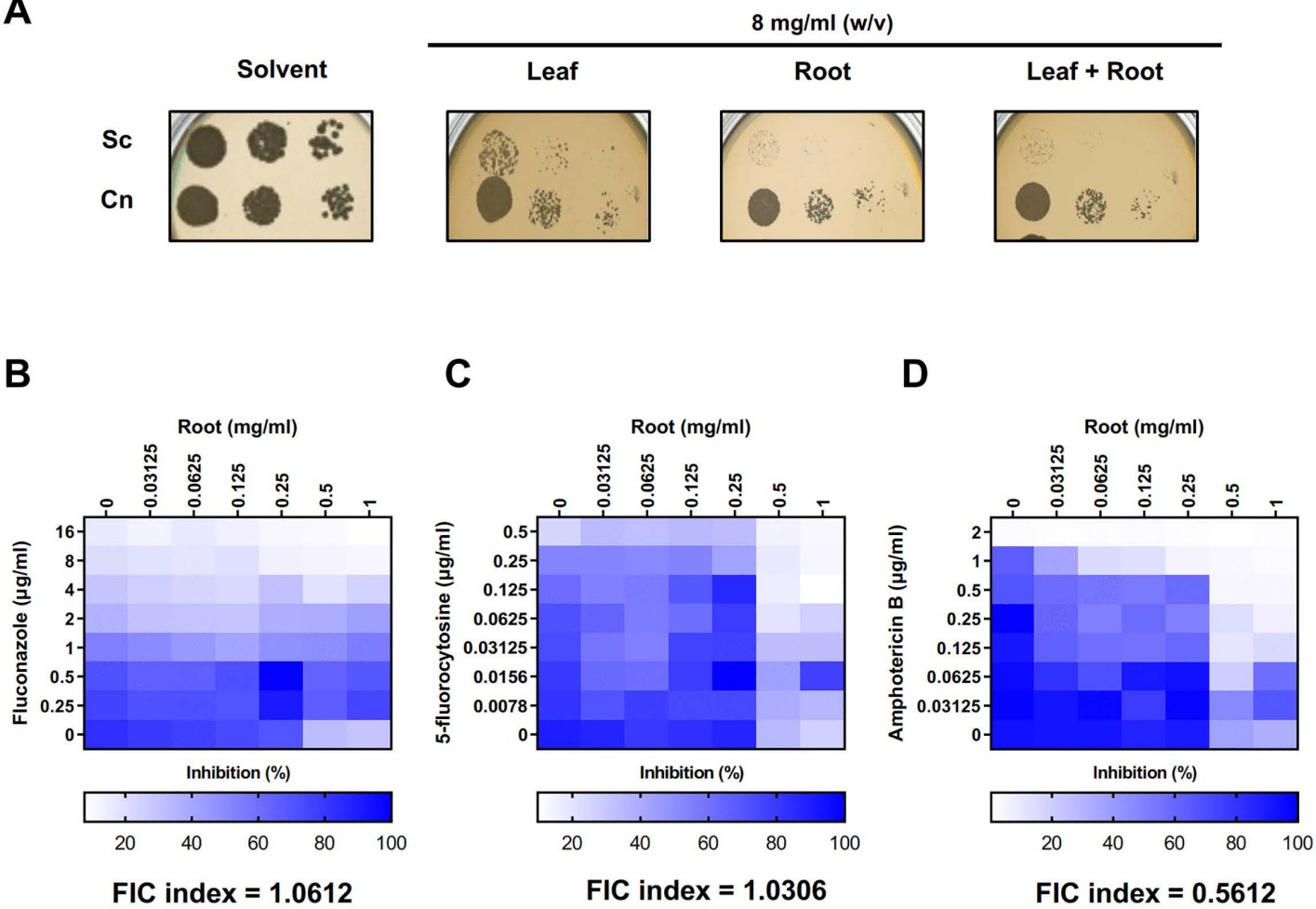

**Fig 2. Antifungal activity of *R. nasutus* extract on yeast growth (A) Spot Assay with two fungal species (*S. cerevisiae* [Sc], and *C. neoformans* [Cn]) was performed on four conditions (YPD medium with solvent, leaf, root, and combined leaf-root).** Concentrations of extracts were diluted to a concentration of 8 mg/mL (w/v), calculated based on the plant dry weight per final volume. Strong fungal growth inhibition was observed, particularly with the root extract, against *S. cerevisiae* and *C. neoformans*. **(B–D)** Checkerboard assays showing the inhibition percentages of *R. nasutus* root extract in combination with antifungal drugs: **(B)** fluconazole, **(C)** 5-fluorocytosine, and amphotericin **B.**

To further investigate the potential of *R. nasutus* extract, we examined its interaction with antifungal drugs using synergistic checkerboard assays. We tested three combinations: *R. nasutus* extract with fluconazole, 5-fluorocytosine, and amphotericin B (Fig 2B–D). Our findings indicated that all three conditions generally displayed indifference. The combinations with fluconazole (FIC index = 1.0612) and 5-fluorocytosine (FIC index = 1.0306) displayed similar additive effects, with no significant synergy observed (Fig 2B and D). In contrast, the combination with amphotericin B demonstrated a near-synergistic effect, with an FIC index of 0.5612 (Fig 2C), suggesting the potential for therapeutic enhancement when paired with the root extract [29,30]. These findings highlight the varying degrees of interaction between *R. nasutus* extract and different antifungal agents, suggesting that its combined use with amphotericin B could be further explored for therapeutic applications.

These results indicate the potential of *R. nasutus* extract as an adjuvant therapy when combined with amphotericin B, while other combinations may require further optimization. Next, we focused on the effect of *R. nasutus* root extract on the virulence factors of *C. neoformans*, including melanin production, capsule formation, and cell morphology, all of which are critical to its pathogenicity [35–39].

### *R. nasutus* extract inhibits melanin production, a key virulence factor

The ability of *C. neoformans* to produce melanin was evaluated by growing the yeast on minimal medium (MM) with L-DOPA, a melanin precursor [31,38]. In untreated and solvent-treated control samples, dark pigmentation indicative of melanin production was observed (Fig 3A and S4 Fig). However, *C. neoformans* treated with the root extract displayed markedly lighter pigmentation, indicating that the root extract interferes with melanin biosynthesis. Since melanin is critical for protecting *C. neoformans* against host immune defenses, this inhibition may weaken the virulence of the fungus and ability to survive in certain environments.

### Impairment of capsule formation as a virulence factor by *R. nasutus* root extract

The effect of *R. nasutus* root extract on capsule formation was next examined by culturing *C. neoformans* in Sabouraud dextrose broth (SDB) [33,34,39]. Capsule formation was noticeably reduced in root extract-treated cultures after 3 and 5 days, compared to the untreated and solvent-treated controls (Fig 3B and S3A Fig). This suggests that the root extract disrupts the ability of *C. neoformans* to form its protective capsule, a key virulence factor for survival in the host. Further quantitative analysis of capsule size was performed to assess the impact of root extract on capsule formation. Scatter plots showed a significant reduction in capsule size in the root extract-treated cultures (Fig 3C and S3B Fig). The data points for the root extract treatment consistently demonstrate smaller capsule sizes compared to the solvent-treated controls, strongly supporting the inhibitory effect of *R. nasutus* root extract on capsule formation in *C. neoformans*.

### Abnormalities in cell morphology of *C. neoformans* induced by *R. nasutus* root extract

The impact of *R. nasutus* root extract on the cell morphology of *C. neoformans* was assessed microscopically (Fig 4A–D). Surprisingly, treated cells exhibited noticeable abnormalities compared to untreated (normal) cells in both YPD and SDB media, with the effects being more pronounced in the latter. These abnormalities included irregular shapes, reduced cell size, and alterations in internal structures. The nature and underlying causes of these abnormalities remain unknown and warrant further investigation.

Significant changes in cell appearance were evident, highlighting the impact of the extract on cellular integrity. These findings suggest that *R. nasutus* root extract has a strong effect on the overall morphology of *C. neoformans*, potentially compromising its ability to maintain normal cellular integrity.

## Discussion

This study demonstrates the antifungal potential of *R. nasutus*, a plant long revered in traditional Thai medicine, with particular activities against *C. neoformans*, a major fungal pathogen associated with high mortality and limited treatment

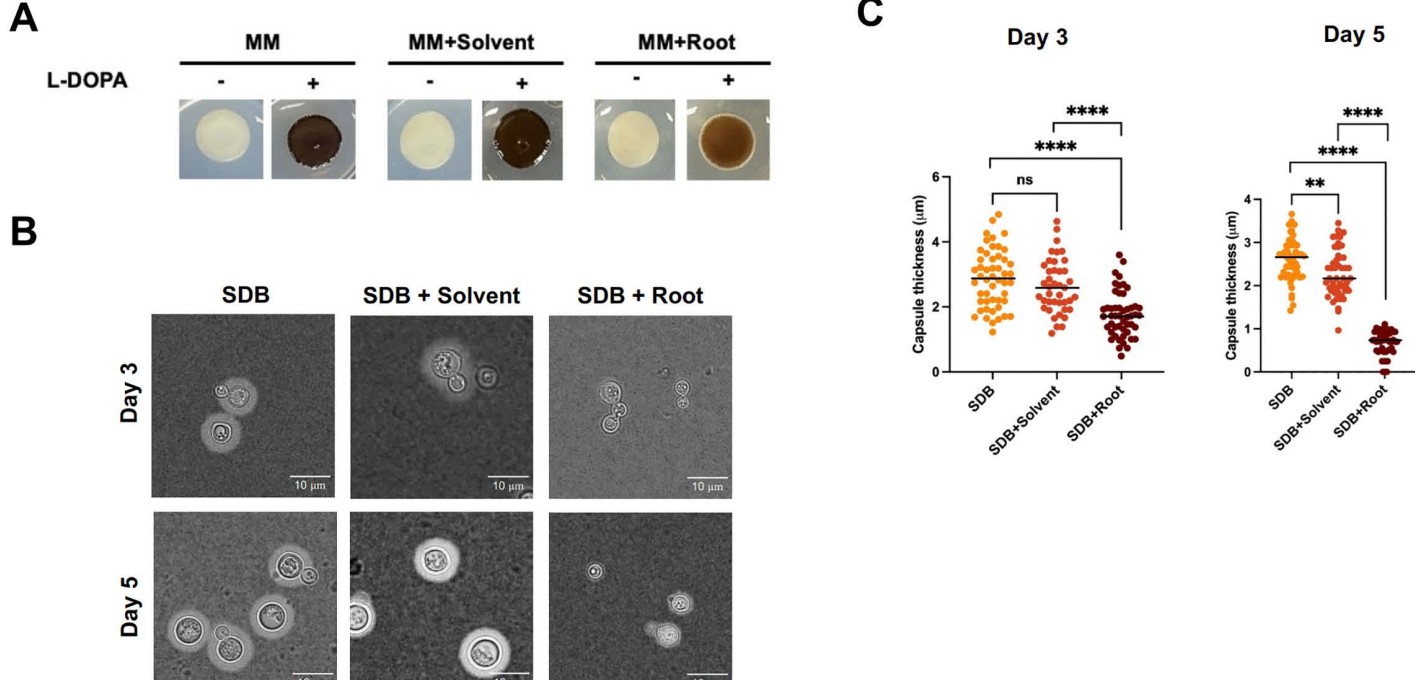

**Fig 3. Effect of root extract on the virulence factors of *C. neoformans*.** All assays were performed using the same extract concentration of 8 mg/mL (w/v), calculated based on the plant dry weight per final volume. **(A)** The melanin production in *C. neoformans* grown on minimal medium (MM) with and without L-DOPA, a precursor for melanin synthesis. Control conditions (MM alone and MM with solvent) exhibited dark pigmentation indicative of melanin production, whereas the addition of *R. nasutus* root extract significantly reduced melanin synthesis, resulting in lighter pigmentation. **(B)** Microscopic image of capsule formation in *C. neoformans* grown in tenfold-diluted Sabouraud dextrose broth (SDB) with 50 mM MOPS buffer. Capsule formation was significantly reduced in the presence of *R. nasutus* root extract compared to the solvent control and untreated conditions at days 3 and 5. **(C)** Quantification of capsule size from Fig 3B. Scatter plots show a significant reduction in capsule size in *C. neoformans* treated with root extract compared to the solvent control and untreated samples. Statistical analysis was carried out by unpaired t-test ($n = 50$ per condition; **, $P < 0.01$; ****, $P < 0.0001$; ns, not significant).

options. Among the tested yeast species, the extract exhibited the strongest inhibitory effects on *C. neoformans* and *S. cerevisiae*, highlighting the potential of *R. nasutus* as a candidate for further antifungal development.

Previous investigations of *R. nasutus* have reported broad antimicrobial activity, including antifungal effects against *Candida albicans* and *Aspergillus* spp. [11,13,24,25,40]. However, most of these studies focused primarily on growth inhibition. In contrast, our findings provide new insights by showing that the root extract not only suppresses fungal growth but also interferes with virulence traits in *C. neoformans*. Specifically, we observed significant reductions in melanin production and capsule formation—two hallmarks of cryptococcal pathogenesis that, to our knowledge, have not previously been studied in relation to *R. nasutus*. Because melanin protects fungal cells against oxidative stress and immune attack, and the capsule shields them from phagocytosis, inhibition of both traits may substantially weaken the pathogen's ability to cause disease.

In addition to attenuating these virulence traits, the root extract induced notable morphological abnormalities in *C. neoformans* cells, including changes in size and surface integrity. Such morphological defects—altered cell size, disturbed surface integrity—are consistent with previous observations that perturbations to cell-wall composition in *C. neoformans* impair cell-wall integrity, morphology, and viability [41–44]. Preliminary phytochemical analysis suggests that rhinacanthin-C is likely the major active compounds present in the root extract. While absolute quantification was not

 

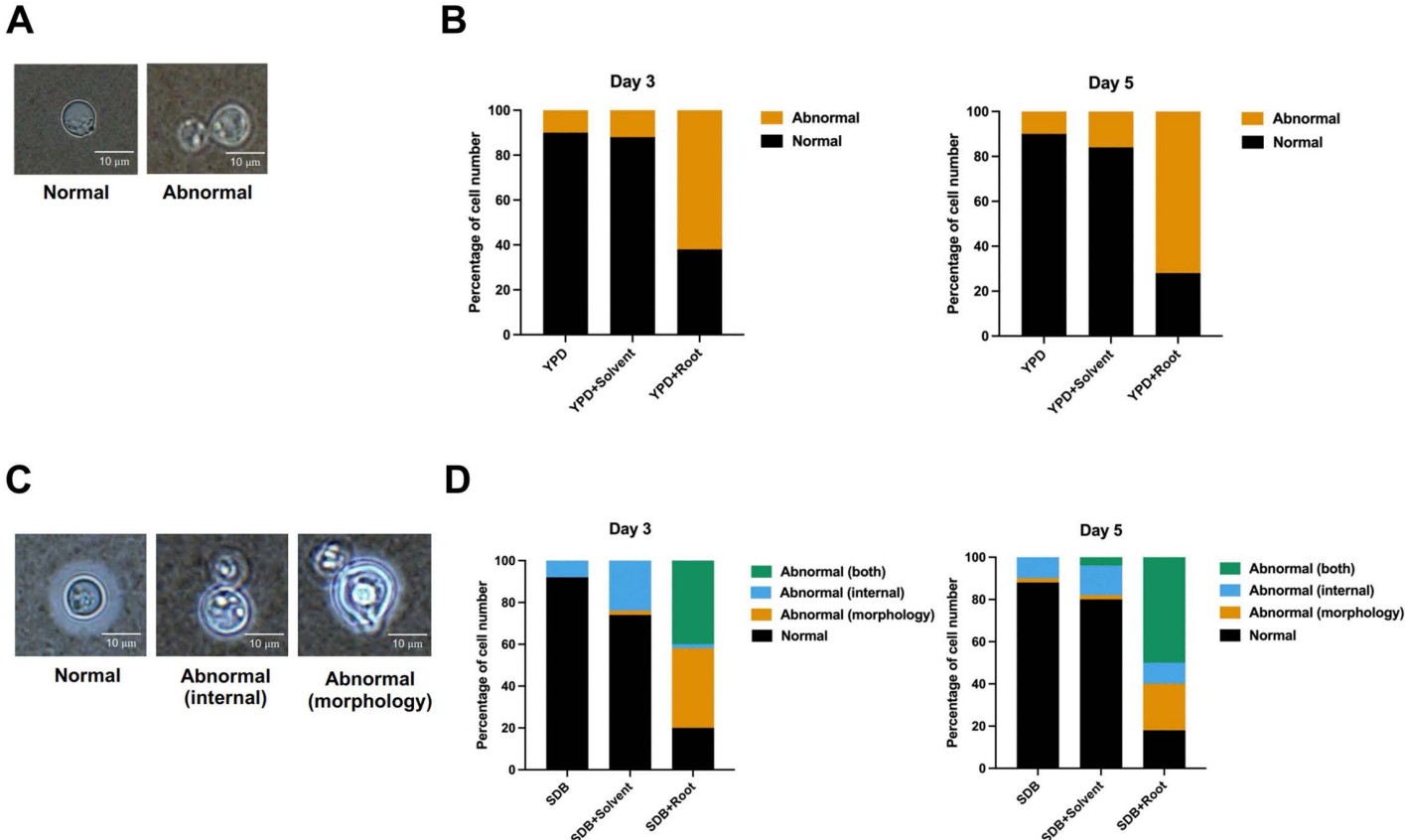

**Fig 4. Abnormalities in cell morphology of *C. neoformans* due to root extract.** All assays were performed using the same extract concentration of 8 mg/mL (w/v), calculated based on the plant dry weight per final volume. **(A)** Microscopic image of normal and abnormal *C. neoformans* cells in YPD media. **(B)** Graph showing the distribution of normal and abnormal *C. neoformans* cells cultured in YPD media. A greater number of abnormal cells were found in YPD media with root extract. **(C)** Microscopic image of normal and abnormal *C. neoformans* cells in SDB media. Abnormalities are classified as either internal (internal structural defects) or external (alterations to the morphology of the cell). **(D)** Graph showing distribution of normal, internal abnormal, and external abnormal *C. neoformans* cells cultured in SDB media. Cells grown in SDB media with root extract, showed the largest proportion of cells with both internal and external abnormalities. Data represent analysis of 50 cells per condition (*n* = 50).

possible due to the lack of commercial standards, our estimates based on published methods suggest that rhinacanthin-C is present at higher concentrations in the root extract than in the leaf extract, making the root preparation the most promising candidate for antifungal applications [24,25]. Further work is needed to isolate rhinacanthin-C and evaluate its individual contribution to the observed effects. It should be noted, however, that all biological effects reported in this study were observed using the crude extract, which contains multiple phytochemical constituents. Although rhinacanthin-C is the most abundant compound detected, other components in the extract may also contribute to the antifungal and antivirulence activities observed.

Another important aspect of our study is the interaction of *R. nasutus* extract with standard antifungal drugs. Several studies have reported synergistic interactions between conventional antifungal agents and plant-derived extracts against *C. neoformans*. For example, the ethyl acetate fraction (EAF) obtained from the stem bark of *Poincianella pluviosa* exhibited synergistic fungicidal activity with amphotericin B (AMB), with fractional inhibitory concentration (FICI) values ranging from 0.03 to 0.06 [45]. In another study, sclareolide, a purified compound derived from *Salvia sclarea*, was shown to reduce the MIC of AMB by several folds, although the reported FICI values were >1 and generally interpreted as an

indifferent interaction [46]. Encouragingly, our *R. nasutus* root extract showed near-synergistic antifungal activity when combined with amphotericin B, a drug combination commonly used in the management of cryptococcal meningitis [47]. These findings suggest that *R. nasutus* root extract could serve as an adjuvant therapy to enhance current treatment efficacy and potentially reduce drug dosages, thereby minimizing side effects and resistance development.

Despite these promising findings, our study has limitations. All experiments were conducted in vitro, and fungal cell-wall properties are known to change dynamically in response to environmental or host-associated stresses. Indeed, in C. neoformans, both culture conditions and host infection environments significantly influence cell-wall composition, architecture, and surface exposure of cell-wall components, which in turn modulate cell morphology, immune recognition and virulence. Therefore, the in vivo efficacy of *R. nasutus* extracts may differ from our in vitro observations. The molecular targets and pharmacokinetic properties of the active components also remain unknown. Future studies should thus focus on isolating and characterizing the bioactive constituents, clarifying their mechanisms of action, and evaluating efficacy in animal models. Such studies will be essential before any clinical application can be considered.

In summary, our findings provide a scientific basis for the traditional use of *R. nasutus* and identify its root extract as a promising source of antifungal activity against *C. neoformans*. By inhibiting key virulence factors and enhancing the activity of existing antifungal drugs, *R. nasutus* offers potential as both a direct antifungal agent and an adjuvant to improve current treatment strategies.

## Supporting information

**S1 Fig. Combined HPLC chromatograms of *R. nasutus* extracts.** Overlay of HPLC chromatograms from leaf extract, root extract, and combined leaf–root extract, presented on a unified Y-axis to facilitate direct comparison of their chemical profiles.
(TIF)

**S2 Fig. Antifungal activity of *R. nasutus* extracts at different concentrations.** Spot assays with two fungal species (*S. cerevisiae* [Sc] and *C. neoformans* [Cn]) were performed under four conditions: YPD medium supplemented with solvent (control), leaf extract, root extract, and combined leaf + root extracts. Extract concentrations (4, 8, and 16 mg/mL) were calculated based on the dry weight of plant material relative to the final volume. The strongest antifungal effect was observed at 8 mg/mL, particularly with the root and leaf + root extracts, suggesting a non-linear concentration–response relationship against both fungal species. Colony images of *C. neoformans* were analyzed using ImageJ 1.53t: Colony images of *C. neoformans* were quantified using ImageJ 1.53t, and growth was expressed as the area under the curve (AUC) relative to the solvent control. Data points showing marked reductions in growth compared with the control are highlighted in bold red.
(TIF)

**S3 Fig. Melanin production in *C. neoformans* treated with root extract.** Melanin production, corresponding to Fig 3A, was quantified in cells exposed to the root extract, with solvent-treated cells as the control. Pixel intensity values (0–255 gray levels) were measured and presented as histograms. Details of the quantification method are provided in the Materials and methods section.
(TIF)

**S4 Fig. Effect of root extract on *C. neoformans* cell size.** (A) Microscopy images showing cell morphology of *C. neoformans* treated with root extract at 8 mg/mL (calculated as plant dry weight per final volume). Cells were stained with India ink and showed smaller sizes compared with untreated cells in YPD medium at both Day 3 and Day 5. (B) Scatter plots of quantified measurements demonstrate a significant reduction in cell size in root extract–treated samples compared to solvent control and untreated groups. Cell and capsule sizes were measured from $n = 50$ cells per condition.
(TIF)

**S1 Table. Yeast strains used in this study.**
(XLSX)

**S1 File. Supporting information (raw data).**
(XLSX)

## Acknowledgments

We thank Prof. Dr. Paweena Traiperm, Department of Plant Science, Faculty of Science, Mahidol University, for her taxonomic authentication of *R. nasutus*. We also thank Chanakan Techawisutthinan for her technical assistance and for contributing to the discussion. The following reagent was obtained through BEI Resources, NIAID, NIH as part of the Human Microbiome Project: *C. neoformans* KN99α (NR-48769). We would also like to thank the herb farm, especially Sangat Phrommes, for their support in providing the *R. nasutus* plants for our project, as well as CIF personnel, particularly Sirichai Kositarat and Pradup Mesawat, for their assistance with the HPLC analysis.

During the preparation of this work, the authors used ChatGPT-4o and ChatGPT-5 to assist with improving readability and clarity. All content generated with the assistance of this tool was subsequently reviewed and edited by the authors, who take full responsibility for the final content of the published article.

## Author contributions

**Conceptualization:** Sittinan Chanarat.

**Formal analysis:** Muthita Khongthongdam, Trinity Jantarach, Tanaporn Phetruen, Sittinan Chanarat.

**Funding acquisition:** Sittinan Chanarat.

**Investigation:** Muthita Khongthongdam, Trinity Jantarach, Tanaporn Phetruen.

**Methodology:** Muthita Khongthongdam, Trinity Jantarach, Tanaporn Phetruen, Sittinan Chanarat.

**Resources:** Sittinan Chanarat.

**Supervision:** Sittinan Chanarat.

**Writing – original draft:** Muthita Khongthongdam, Trinity Jantarach, Tanaporn Phetruen, Sittinan Chanarat.

**Writing – review & editing:** Muthita Khongthongdam, Trinity Jantarach, Tanaporn Phetruen, Sittinan Chanarat.

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
