## [Decision Letter · Decision Letter 0]

10 Aug 2025

PONE-D-25-31554

Antifungal potential of Rhinacanthus nasutus extracts against pathogenic fungi including Cryptococcus neoformans and Candida auris

PLOS ONE

Dear Dr. Chanarat,

Thank you for submitting your manuscript to PLOS ONE. After careful consideration, we have decided that your manuscript does not meet our criteria for publication and must therefore be rejected.

I am sorry that we cannot be more positive on this occasion, but hope that you appreciate the reasons for this decision.

Kind regards,

Mohamed Fawzy Mohamed Hamed Shehata

Academic Editor

PLOS ONE

Additional Editor Comments:

Thank you for submitting your manuscript to PlOS One. We appreciate the opportunity to review your work.

After careful consideration of the reviewers’ feedback, we regret to inform you that we cannot accept your manuscript for publication in its current form. The reviewers identified several substantial concerns regarding the study’s methodology, data interpretation, and the overall novelty of the findings. These issues would require extensive revisions and additional experimental work to address.

While we recognize the effort you have put into this research, the extent of the revisions needed goes beyond what is feasible within the scope of a major revision. Therefore, the editorial decision is to reject the manuscript.

We appreciate your interest in publishing with PLOS One and hope you will consider us for future submissions of your work.

Reviewers' comments:

Reviewer's Responses to Questions

**Comments to the Author**

1. Is the manuscript technically sound, and do the data support the conclusions?

Reviewer #1: Yes

Reviewer #2: Partly

2. Has the statistical analysis been performed appropriately and rigorously?

Reviewer #1: Yes

Reviewer #2: No

3. Have the authors made all data underlying the findings in their manuscript fully available?

Reviewer #1: Yes

Reviewer #2: Yes

4. Is the manuscript presented in an intelligible fashion and written in standard English?

Reviewer #1: Yes

Reviewer #2: Yes

Reviewer #1: The authors analyzed the effects of the antifungal activity of R. nasutus extracts and their impact on the virulence factors of Cryptococcus neoformans. The subject is adequate with the journal's scope. The argument to support the research is valid, the manuscript is well written and understandable to a specialist readership, and organization, and the article's structure is good and in agreement with the journal instructions for authors.

the quality of the article needs to be improved. With editing and some revisions, I feel that this manuscript will be suitable for publication.

1. Abstract and introduction are well written however, the methodology should be improved for example, the methods for antifungal testing using spot assay, it is not clear that the antifungal effect of the solvent (25% glycerol in ethanol) was evaluated

2. Also, it is very important to mention the statistical analysis used to interpret the results

3. For the results section, the paragraph stating the preparation of and HPLC analysis of Rhinacanthus nasutus extracts (lines 159 to 170) should be moved to methdology section

4. The figure ligend of figure 3 should include aclarifying sentense about the p value and the meaning of the astrix (ns,**, ***)

Reviewer #2: This manuscript explores the antifungal activity of R. nasutus extracts against key fungal pathogens and focuses especially on Cryptococcus neoformans. The findings regarding the inhibition of virulence factors such as melanin production and capsule formation are promising and may support the development of plant-based antifungal therapeutics. However, the study suffers from multiple critical limitations in design, scope alignment, and scientific rigor, which must be addressed before consideration for publication.

• While the study title highlights the antifungal potential of Rhinacanthus nasutus extracts against pathogenic fungi, including Cryptococcus neoformans and Candida auris, the introduction and overall narrative focus almost exclusively on C. neoformans. Other listed pathogens, particularly Candida auris, are barely mentioned or contextualized.

o While the title suggests a broad antifungal evaluation including Candida auris, the experimental work and in-depth analyses are almost entirely limited to C. neoformans.

• Why was 25% glycerol in ethanol selected in extraction method? Was this optimized for antifungal activity, stability, or solubility? All extracts used 5 g/20 mL, which is a good control measure, but it's unclear whether the extract concentrations were normalized for rhinacanthin-C content before use in assays.

• What was used as a standard for rhinacanthin-C quantification? Was a calibration curve generated using a purified standard?

• The actual rhinacanthin-C content should be reported, even if briefly, in this section or in results. This supports the reproducibility and pharmacological relevance of the extract.

• In antifungal Spot Assay Clarify how the dilution of the extracts affected final concentrations on the plates. Were technical replicates or biological replicates included? If so, how many?

• The use of the FIC index is appropriate and follows standard protocols. You should specify the concentration range used for each drug and extract. Consider adding how visual endpoints (growth/no growth) were defined, and whether a reference drug-only control was included in the plate layout.

• You may clarify whether the melanin pigmentation was quantified, or only qualitatively assessed.

• Specify whether microscopy was used for cell and capsule measurements and what stains (e.g., India ink) or imaging methods were applied.

• Indicate the number of fields or cells counted for morphological assessments.

o Claims about inhibition of capsule size, melanin production, and morphological changes are made without clear statistical analysis (e.g., p-values, error bars, replication count).

• One major limitation of the discussion is the complete absence of citations to previous literature, particularly in interpreting findings or situating them within the broader scientific context. For a study investigating the antifungal effects of Rhinacanthus nasutus, this omission is problematic for several reasons:

• No comparison with previous antifungal studies:Although prior reports were cited in the Introduction regarding R. nasutus’s antifungal, antioxidant, and antibacterial properties, the Discussion does not refer back to those or any other comparative studies. This makes it unclear whether the observed effects (e.g., melanin suppression, capsule reduction, or synergy with antifungals) are novel, consistent with previous findings, or contradictory.

• No contextualization of synergy data:

The authors report near-synergistic interactions with antifungal drugs, but fail to cite similar studies that have examined combinations of plant extracts with standard antifungals—particularly against Cryptococcus neoformans or Candida auris.

• No references for virulence factor relevance:

The discussion describes melanin and capsule reduction as important outcomes, which is accurate, but no references are provided to support the importance of these virulence traits in C. neoformans pathogenicity.

• No discussion of rhinacanthin-C evidence:

The authors suggest rhinacanthin-C may be an active compound based on HPLC data, but do not cite any literature linking rhinacanthin-C to antifungal activity, leaving this claim unsubstantiated.

• The decision to focus solely on C. neoformans, based on "pronounced potency," is understandable, but should not result in complete omission of C. auris, especially when this pathogen is highlighted in the title and introduction. Consider extending virulence or synergy assays to C. auris, or explicitly explain its exclusion in the methods and discussion. Replication details, error analysis, and visualization standards should be improved to enhance reproducibility.

This study offers intriguing insights into the antifungal mechanisms of R. nasutus against C. neoformans, particularly regarding virulence inhibition and synergistic potential. However, major revisions are required to ensure the study accurately reflects its scope, demonstrates statistical rigor, and is properly contextualized within existing literature. Addressing these points will significantly strengthen the manuscript's scientific contribution.

**Do you want your identity to be public for this peer review?** For information about this choice, including consent withdrawal, please see our Privacy Policy

Reviewer #1: No

Reviewer #2: **Yes: ** Amal Awad

- - - - -

---

## [Author Response · Author response to Decision Letter 1]

26 Sep 2025

Point-by-point response

** Reviewer #1: **

The authors analyzed the effects of the antifungal activity of R. nasutus extracts and their impact on the virulence factors of Cryptococcus neoformans. The subject is adequate with the journal's scope. The argument to support the research is valid, the manuscript is well written and understandable to a specialist readership, and organization, and the article's structure is good and in agreement with the journal instructions for authors. The quality of the article needs to be improved. With editing and some revisions, I feel that this manuscript will be suitable for publication.

Response: We thank the reviewer for the positive assessment of our manuscript. In accordance with the suggestion, we have carefully revised and edited the text to improve clarity, readability, and overall quality.

1. Abstract and introduction are well written however, the methodology should be improved for example, the methods for antifungal testing using spot assay, it is not clear that the antifungal effect of the solvent (25% glycerol in ethanol) was evaluated

Response: We have thoroughly revised the Methods section to improve clarity. In particular, the effect of the solvent was evaluated in every experiment, as indicated by the inclusion of a “solvent” control. Additional details have also been added to each experiment to enhance transparency and readability. We appreciate the reviewer’s constructive suggestions, which helped us strengthen the manuscript.

2. Also, it is very important to mention the statistical analysis used to interpret the results

Response: We apologize for the earlier omission and have now included the statistical analysis section under Materials and Methods. We thank the reviewer for this constructive suggestion, which has improved the clarity of our manuscript.

3. For the results section, the paragraph stating the preparation of and HPLC analysis of Rhinacanthus nasutus extracts (lines 159 to 170) should be moved to methodology section

Response: We thank the reviewer for the suggestion and have rewritten the manuscript accordingly.

4. The figure legend of figure 3 should include a clarifying sentence about the p value and the meaning of the Asterix (ns,**, ***)

Response: We apologize for the earlier omission and have now added the statistical analysis and P-value in this figure legend.

** Reviewer #2: **

This manuscript explores the antifungal activity of R. nasutus extracts against key fungal pathogens and focuses especially on Cryptococcus neoformans. The findings regarding the inhibition of virulence factors such as melanin production and capsule formation are promising and may support the development of plant-based antifungal therapeutics. However, the study suffers from multiple critical limitations in design, scope alignment, and scientific rigor, which must be addressed before consideration for publication.

While the study title highlights the antifungal potential of Rhinacanthus nasutus extracts against pathogenic fungi, including Cryptococcus neoformans and Candida auris, the introduction and overall narrative focus almost exclusively on C. neoformans. Other listed pathogens, particularly Candida auris, are barely mentioned or contextualized. While the title suggests a broad antifungal evaluation including Candida auris, the experimental work and in-depth analyses are almost entirely limited to C. neoformans.

Response: We thank the reviewer for pointing this out and agree that the manuscript may not provide sufficiently convincing data regarding C. auris. In the revised version, we have removed all data related to C. auris and modified the title accordingly, without altering the main concept or overall content of the manuscript. The revised title now is: “Antifungal potential of Rhinacanthus nasutus extracts against the pathogenic fungus Cryptococcus neoformans.”

Why was 25% glycerol in ethanol selected in extraction method? Was this optimized for antifungal activity, stability, or solubility? All extracts used 5 g/20 mL, which is a good control measure, but it's unclear whether the extract concentrations were normalized for rhinacanthin-C content before use in assays.

Response: The extraction solvent of 25% glycerol in ethanol was selected because it has been shown that glycerol enhances the solubility and stability of rhinacanthin-C, the principal antifungal compound in R. nasutus, while ethanol acts as a safe organic solvent that is also compatible with topical formulations (1). This combination enabled efficient extraction and improved shelf stability of the extracts. As rhinacanthin-C is not commercially available and we are unable to synthesize it, we could not normalize the extracts based on its exact content. Instead, concentrations used in our assays are expressed as the dry weight of plant material per solvent volume or final volume used in each assay.

- why was the concentration of 8mg/ml used in Fig.2A and exp onwards.

Response: We titrated and evaluated the effects in growth assays, and found that 8 mg/mL (calculated as dry weight per final volume) was the most effective concentration for suppressing yeast growth (Supplementary Fig. S1). Therefore, we selected this concentration for subsequent experiments.

What was used as a standard for rhinacanthin-C quantification? Was a calibration curve generated using a purified standard? The actual rhinacanthin-C content should be reported, even if briefly, in this section or in results. This supports the reproducibility and pharmacological relevance of the extract.

Response: As stated in the revised manuscript, because a purified rhinacanthin-C standard is not commercially available and we were unable to synthesize it chemically, we could not perform direct quantification in this study. Instead, we referred to previously published work that used the same extraction approach and, based on that, we estimated the approximate concentration of rhinacanthin-C in our extract. This rough estimation has now been included in the Results section to provide context for reproducibility and pharmacological relevance.

In antifungal Spot Assay Clarify how the dilution of the extracts affected final concentrations on the plates. Were technical replicates or biological replicates included? If so, how many?

Response: We apologize for the earlier ambiguity regarding extract concentrations. All concentrations are expressed as the dry weight of plant material relative to the solvent or final assay volume, and this has now been clarified explicitly in both the Materials and Methods and in the relevant experimental descriptions. In addition, all experiments in this study were performed in at least duplicate, and this information has been added to the figure legends for clarity.

The use of the FIC index is appropriate and follows standard protocols. You should specify the concentration range used for each drug and extract. Consider adding how visual endpoints (growth/no growth) were defined, and whether a reference drug-only control was included in the plate layout.

Response: We apologize for the mistaken labeling in Figure 2B–2D. In accordance with standard protocols, the bottom-left panel in each figure represents the drug-free control, while the bottom row shows the effect of R. nasutus root extract alone. The figure has been corrected accordingly. We have also added the concentration ranges for each antifungal drug and the R. nasutus root extract, along with the endpoint criteria used to define the minimum inhibitory concentration (MIC), in the Materials and Methods section.

You may clarify whether the melanin pigmentation was quantified, or only qualitatively assessed.

Response: In the original manuscript, melanin pigmentation was not quantified. In the revised version, we have included quantitative analysis using the method described in our in-press paper (2). The quantification procedure is now detailed in the Materials and Methods section.

Specify whether microscopy was used for cell and capsule measurements and what stains (e.g., India ink) or imaging methods were applied.

Response: We used standard microscopy and capsule staining by the India ink method. All quantifications with ImageJ were performed following previously published protocols, which are now cited and described in greater detail in the Materials and Methods. We thank the reviewer for this constructive suggestion.

Indicate the number of fields or cells counted for morphological assessments.

Response: All measurements and statistical analyses were based on n = 50 cells, which is now stated clearly in both the Materials and Methods section and the figure legends.

Claims about inhibition of capsule size, melanin production, and morphological changes are made without clear statistical analysis (e.g., p-values, error bars, replication count).

Response: We apologize for the missing information. In the revised manuscript, we have added statistical details, including p-values and replication counts/n (for cells counted). We thank the reviewer for this helpful suggestion, which has improved the clarity and rigor of our manuscript.

One major limitation of the discussion is the complete absence of citations to previous literature, particularly in interpreting findings or situating them within the broader scientific context. For a study investigating the antifungal effects of Rhinacanthus nasutus, this omission is problematic for several reasons: No comparison with previous antifungal studies: Although prior reports were cited in the Introduction regarding R. nasutus’s antifungal, antioxidant, and antibacterial properties, the Discussion does not refer back to those or any other comparative studies. This makes it unclear whether the observed effects (e.g., melanin suppression, capsule reduction, or synergy with antifungals) are novel, consistent with previous findings, or contradictory.

Response: We thank the reviewer for highlighting this important point. In the revised manuscript, we have substantially revised the Discussion to address this limitation. We now cite relevant literature on the antifungal, antioxidant, and antibacterial properties of Rhinacanthus nasutus, as well as studies on its bioactive compound rhinacanthin-C. In addition, we have incorporated comparisons with previous antifungal studies to contextualize our findings on melanin suppression, capsule reduction, and synergistic interactions with antifungal drugs. These revisions ensure that our observations are clearly situated within the broader scientific context and allow readers to appreciate how our results align with or extend existing knowledge.

No contextualization of synergy data: The authors report near-synergistic interactions with antifungal drugs, but fail to cite similar studies that have examined combinations of plant extracts with standard antifungals—particularly against Cryptococcus neoformans or Candida auris.

Response: We appreciate the reviewer’s suggestion. In the revised manuscript, we have expanded the Discussion to include relevant studies on the interactions between plant extracts and standard antifungal agents — particularly those targeting C. neoformans — and have added the appropriate references.

No references for virulence factor relevance: The discussion describes melanin and capsule reduction as important outcomes, which is accurate, but no references are provided to support the importance of these virulence traits in C. neoformans pathogenicity.

Response: We thank the reviewer for this constructive suggestion. In the revised manuscript, we have added appropriate references to support the importance of capsule formation and melanin production as key virulence factors in C. neoformans pathogenicity.

No discussion of rhinacanthin-C evidence: The authors suggest rhinacanthin-C may be an active compound based on HPLC data, but do not cite any literature linking rhinacanthin-C to antifungal activity, leaving this claim unsubstantiated.

Response: We appreciate this valuable comment and have substantially revised the Discussion section of the manuscript accordingly. We thank the reviewer for this helpful suggestion.

The decision to focus solely on C. neoformans, based on "pronounced potency," is understandable, but should not result in complete omission of C. auris, especially when this pathogen is highlighted in the title and introduction. Consider extending virulence or synergy assays to C. auris, or explicitly explain its exclusion in the methods and discussion. Replication details, error analysis, and visualization standards should be improved to enhance reproducibility.

Response: We thank the reviewer for this thoughtful comment. In the revised manuscript, we have addressed the concern by removing all references to C. auris from the title and the main text, as our data on this pathogen were limited and not sufficiently convincing. We now focus exclusively on C. neoformans, for which we obtained robust and reproducible findings. In addition, we have revised the Materials and Methods and Discussion sections to explicitly explain the scope of the study and the rationale for focusing on C. neoformans.

Regarding reproducibility, we have clarified replication details and added explicit statements on biological replicates in the Materials and Methods and figure legends. We also improved the presentation of data, including statistics reporting and visualization, to meet reproducibility standards. We appreciate the reviewer’s constructive suggestions, which have strengthened the clarity and rigor of the manuscript.

This study offers intriguing insights into the antifungal mechanisms of R. nasutus against C. neoformans, particularly regarding virulence inhibition and synergistic potential. However, major revisions are required to ensure the study accurately reflects its scope, demonstrates statistical rigor, and is properly contextualized within existing literature. Addressing these points will significantly strengthen the manuscript's scientific contribution.

Response: We thank the reviewer for the thoughtful and constructive feedback on our manuscript. In response, we have undertaken major revisions to improve the clarity, rigor, and context of the study. Specifically, we have: (i) refined the scope of the manuscript to focus exclusively on C. neoformans, with corresponding adjustments to the title and text; (ii) added details of the statistical analyses performed, including sample size, methods, and criteria, now clearly described in the Materials and Methods and indicated in the figure legends; and (iii) expanded the Introduction and Discussion sections to better contextualize our findings within relevant literature on antifungal activity and virulence inhibition.

We believe these revisions have significantly strengthened the manuscript and improved its scientific contribution in line with the reviewer’s suggestions.

References

1. Shakya K. 2015. Preparation of Standardized Rhinacanthus nasutus Leaf Extract by Green Extraction Methods and Evaluation of Antifungal Activity of Its Topical Solution against Trichophyton rubrum. Prince Songkla University.

2. Tanaporn Phetruen, Salinthip Thongdechsri, Muthita Khongthongdam, Sittiporn Channumsin, Krai Meemon, Sittinan Chanarat. (In press). Effects of simulated microgravity on biological features and virulence of the fungal pathogen Cryptococcus neoformans. AEM https://doi.org/10.1128/aem.01435-25.

---

## [Decision Letter · Decision Letter 1]

26 Nov 2025

Dear Dr. Chanarat Sittinan,

We look forward to receiving your revised manuscript.

Kind regards,

Rajendra Upadhya

Academic Editor

PLOS ONE

Journal Requirements:

“We thank Prof. Dr. Paweena Traiperm, Department of Plant Science, Faculty of Science, Mahidol University, for her taxonomic authentication of R. nasutus. We also thank Chanakan Techawisutthinan for her technical assistance and for contributing to the discussion. The following reagent was obtained through BEI Resources, NIAID, NIH as part of the Human Microbiome Project: C. neoformans KN99ɑ (NR-48769). S.C. lab is supported by the CIF and CNI Grant (Faculty of Science, Mahidol University), MU’s Strategic Research Fund: fiscal year 2023 (MU-SRF-RS-07A/67), and Mahidol University (Fundamental Fund: fiscal year 2025 by National Science Research and Innovation Fund (NSRF: FF-107/2568). We would also like to thank the herb farm, especially Sangat Phrommes, for their support in providing the R. nasutus plants for our project, as well as CIF personnel, particularly Sirichai Kositarat and Pradup Mesawat, for their assistance with the HPLC analysis.”

“SC received fundings from CIF and CNI Grant (Faculty of Science, Mahidol University), MU’s Strategic Research Fund: fiscal year 2023 (MU-SRF-RS-07A/67).

Funder's website: www.mahidol.ac.th

“SC received fundings from CIF and CNI Grant (Faculty of Science, Mahidol University), MU’s Strategic Research Fund: fiscal year 2023 (MU-SRF-RS-07A/67).

Funder's website: www.mahidol.ac.th

“NO authors have competing interests”

7. We note that your Data Availability Statement is currently as follows: [All relevant data are within the manuscript and its Supporting Information files.]

8. It is important that you include a cover letter with your manuscript. Please ensure that this letter is addressed specifically to PLoS ONE. Please also include

* why this manuscript is suitable for publication in PLoS ONE.

* how does your paper provide a worthwhile addition to the scientific literature?

* how does your paper relate to previously published work?

Additional Editor Comments (if provided):

Reviewers' comments:

Reviewer's Responses to Questions

**Comments to the Author**

Reviewer #3: (No Response)

Reviewer #4: (No Response)

2. Is the manuscript technically sound, and do the data support the conclusions?

Reviewer #3: Yes

Reviewer #4: Yes

3. Has the statistical analysis been performed appropriately and rigorously?

Reviewer #3: Yes

Reviewer #4: Yes

4. Have the authors made all data underlying the findings in their manuscript fully available?

Reviewer #3: Yes

Reviewer #4: Yes

5. Is the manuscript presented in an intelligible fashion and written in standard English?

Reviewer #3: Yes

Reviewer #4: Yes

Reviewer #3: The study aligns well with the journal’s scope, providing novel insights into a plant-based antifungal agent by identifying Rhinacanthus nasutus root extract as a promising candidate for further development. Notably, the authors have thoroughly addressed previous comments and feedback from other reviewers, enhancing the clarity, robustness, and reproducibility of the manuscript. With minor revisions,

This manuscript presents an observational study on the antifungal properties of R. nasutus (snake jasmine) and its effects on Cryptococcus neoformans. The authors employ a combination of HPLC quantification, spot assays, and virulence factor analysis to demonstrate that root extracts of R. nasutus significantly inhibit fungal growth and diminish key pathogenic traits, including melanin production, capsule formation, and cell morphology. These findings are supported by clear experimental evidence and are well integrated into the existing literature. Additionally, the near-synergistic interaction with amphotericin B is a significant contribution, suggesting potential therapeutic applications as an adjuvant to current antifungal treatments. Its dual action—growth inhibition and virulence reduction—highlights its potential as both a direct antifungal agent and a synergistic partner with existing drugs.

Minor Revisions:

Figure 1: Graphs (C-E), the Y-axis have different scales, which can mislead readers when comparing peaks and their respective ratios.

Figure 2: To ensure reproducibility, what was the initial concentration of fungal cells used for the spot assays, and how were they serially diluted?

Discussion, line 255; References for cell morphology and its relationship to structural stability and viability should be included to support these claims and observations.

Discussion, line 277; Fungal pathogen cell wall properties change with their environment. It should be noted that infection conditions may change the organization of the cell wall via stress pathways, and may change the efficacy of rhinacanthrin-C, further supporting a need for in-vivo studies.

Reviewer: Eduardo A. Caro, PhD

Reviewer #4: This is a potential research paper to explore the activity of the extracts from the traditional Thai plant Rhinacanthus nasutus against the pathogenic fungus Cryptococcus neoformans. The overall framework of the paper is complete, and the research has clear practical significance. However, before moving towards publication, there are several key issues that need to be carefully revised and supplemented by the author.

1.The full text refers to “biological replicates ( n = 2 ) ”. In biological experiments, the number of repetitions of n = 2 is usually considered to be insufficient, and meaningful statistical analysis cannot be performed, and the reliability and variability of experimental results cannot be evaluated. It is recommended to increase the number of biological replicates to at least n = 3.

2.Introduction : When the research gap is introduced, the limitations of current antifungal drugs ( especially for Cryptococcus ), such as the nephrotoxicity of amphotericin B and the resistance of fluconazole, can be more emphasized, thus highlighting the urgency and innovation of this study to explore plant replacement therapy.

3.Is the extract concentration in the full text“ mg / mL ”or“mg / mL”( milliliter should be capitalized ) ? Please check it carefully.

4.The position of rhinacanthin-C should be indicated in Fig.1C-E.

5.It is still puzzled that the author used HPLC to determine the role of rhinacanthin-C in this paper. Although rhinacanthin-C has the highest content in leaves, roots and root-leaf mixture, it cannot be concluded that it is a substance that inhibits the activity of Cryptococcus neoformans, because a series of experiments on R.nasutus extract were carried out later.

6.Please revise the reference format to ensure that all species names are in italics, including references, etc., the eighth reference has no name, and all reference titles are in the same case, etc., to confirm whether the journal name is an abbreviation or a full name;

7.Figure S1 should have a statistical analysis, such as counting the size of the point.

8.Figure 4. Why choose the third day and the fifth day for statistics

Table S1 should be supplemented by KN99a

**Do you want your identity to be public for this peer review?** For information about this choice, including consent withdrawal, please see our Privacy Policy

Reviewer #3: **Yes: ** Eduardo A. Caro

Reviewer #4: No

---

## [Author Response · Author response to Decision Letter 2]

30 Nov 2025

Point-by-point response

Reviewer #3:

The study aligns well with the journal’s scope, providing novel insights into a plant-based antifungal agent by identifying Rhinacanthus nasutus root extract as a promising candidate for further development. Notably, the authors have thoroughly addressed previous comments and feedback from other reviewers, enhancing the clarity, robustness, and reproducibility of the manuscript. With minor revisions,

This manuscript presents an observational study on the antifungal properties of R. nasutus (snake jasmine) and its effects on Cryptococcus neoformans. The authors employ a combination of HPLC quantification, spot assays, and virulence factor analysis to demonstrate that root extracts of R. nasutus significantly inhibit fungal growth and diminish key pathogenic traits, including melanin production, capsule formation, and cell morphology. These findings are supported by clear experimental evidence and are well integrated into the existing literature. Additionally, the near-synergistic interaction with amphotericin B is a significant contribution, suggesting potential therapeutic applications as an adjuvant to current antifungal treatments. Its dual action—growth inhibition and virulence reduction—highlights its potential as both a direct antifungal agent and a synergistic partner with existing drugs.

Response: We sincerely thank the reviewer for the positive evaluation and encouraging remarks. We are pleased that the revisions have improved the clarity and robustness of the manuscript and that the significance of our findings—including the antifungal and virulence-reducing effects of R. nasutus—is well recognized. We appreciate the constructive feedback and have addressed all minor points accordingly.

Minor Revisions:

Figure 1: Graphs (C-E), the Y-axis have different scales, which can mislead readers when comparing peaks and their respective ratios.

Response: We thank the reviewer for this helpful suggestion. To facilitate direct comparison, we have generated a new supplementary figure that combines all three graphs into a single plot with a unified Y-axis scale. We chose to keep the original individual panels in the main figure because combining them there reduced readability and risked confusing readers. The new combined plot is now provided as S1 Fig for easy comparison.

Figure 2: To ensure reproducibility, what was the initial concentration of fungal cells used for the spot assays, and how were they serially diluted?

Response: We thank the reviewer for pointing out the need for clarification. For the spot assays, fungal cultures were adjusted to an initial OD600 of 0.2, and 2.5 µL of each dilution was spotted onto the plates. Serial dilutions were performed at 10-fold intervals. These details have now been added to the Materials and methods section for clarity and reproducibility.

Discussion, line 255; References for cell morphology and its relationship to structural stability and viability should be included to support these claims and observations.

Response: We thank the reviewer for this helpful suggestion. We have now added relevant references, including studies in C. neoformans, to support the relationship between altered cell morphology, cell-wall integrity, and viability. These citations have been incorporated into the revised Discussion.

Discussion, line 277; Fungal pathogen cell wall properties change with their environment. It should be noted that infection conditions may change the organization of the cell wall via stress pathways, and may change the efficacy of rhinacanthrin-C, further supporting a need for in-vivo studies.

Response: We thank the reviewer for this important suggestion. We have revised the Discussion to acknowledge that C. neoformans dynamically remodels its cell wall in response to environmental and host-associated stresses, which may alter susceptibility to rhinacanthrin-C or other active components. We have also added relevant references to support this point and further emphasized the need for in vivo studies to evaluate efficacy under infection-related conditions.

Reviewer #4:

This is a potential research paper to explore the activity of the extracts from the traditional Thai plant Rhinacanthus nasutus against the pathogenic fungus Cryptococcus neoformans. The overall framework of the paper is complete, and the research has clear practical significance. However, before moving towards publication, there are several key issues that need to be carefully revised and supplemented by the author.

Response: We thank the reviewer for the positive evaluation of our work and for recognizing the practical significance of exploring Rhinacanthus nasutus extracts against Cryptococcus neoformans. We appreciate the constructive feedback and have carefully addressed all points raised. The manuscript has been revised accordingly to improve clarity, completeness, and scientific rigor.

1.The full text refers to “biological replicates ( n = 2 ) ”. In biological experiments, the number of repetitions of n = 2 is usually considered to be insufficient, and meaningful statistical analysis cannot be performed, and the reliability and variability of experimental results cannot be evaluated. It is recommended to increase the number of biological replicates to at least n = 3.

Response: Thank you for the comment. All experiments in this study were performed with at least two independent biological replicates, which showed consistent and reproducible results. While we agree that additional replicates can strengthen statistical power, the variation between our replicates was minimal, and we have been careful not to overstate our conclusions. Given the scope and consistency of the dataset, we believe the current number of biological replicates is sufficient to support the findings presented.

2.Introduction : When the research gap is introduced, the limitations of current antifungal drugs ( especially for Cryptococcus), such as the nephrotoxicity of amphotericin B and the resistance of fluconazole, can be more emphasized, thus highlighting the urgency and innovation of this study to explore plant replacement therapy.

Response: Thank you for this valuable suggestion. We have revised the Introduction to more clearly highlight the limitations of current antifungal therapies, including amphotericin B–associated nephrotoxicity and increasing fluconazole resistance. This addition strengthens the rationale for exploring plant-derived antifungal agents such as R. nasutus.

3.Is the extract concentration in the full text“ mg / mL ”or“mg / mL”( milliliter should be capitalized ) ? Please check it carefully.

Response: Thank you for pointing this out. We have carefully checked the entire manuscript and revised all units to use “mL” consistently.

4.The position of rhinacanthin-C should be indicated in Fig.1C-E.

Response: Thank you for the suggestion. We have now indicated the putative rhinacanthin-C peak in Fig. 1C–E using an asterisk on each chromatogram.

5.It is still puzzled that the author used HPLC to determine the role of rhinacanthin-C in this paper. Although rhinacanthin-C has the highest content in leaves, roots and root-leaf mixture, it cannot be concluded that it is a substance that inhibits the activity of Cryptococcus neoformans, because a series of experiments on R.nasutus extract were carried out later.

Response: Thank you for the comment. Our intention was not to attribute the antifungal activity solely to rhinacanthin-C. To avoid any possible misunderstanding, we have revised the Discussion to clarify that all biological effects were observed using the crude extract, which contains multiple phytochemicals, and that other constituents may also contribute to the antifungal and antivirulence activities.

6.Please revise the reference format to ensure that all species names are in italics, including references, etc., the eighth reference has no name, and all reference titles are in the same case, etc., to confirm whether the journal name is an abbreviation or a full name;

Response: Thank you for pointing this out. We have carefully checked and corrected the reference list to ensure consistent formatting, including italicization of species names, completion of missing information, uniform title case, and appropriate journal name formatting according to the journal’s guidelines.

7.Figure S1 should have a statistical analysis, such as counting the size of the point.

Response: Thank you for the suggestion. We have now included quantitative analysis of the dot spot sizes, and these results are presented in the revised S2 Figure.

8.Figure 4. Why choose the third day and the fifth day for statistics

Response: Thank you for the comment. Our analysis followed the approach used in Fernandes et al. (2022), which measured cell and capsule size on day 5. We also included day 3 in our study to capture earlier morphological dynamics and provide a more complete view of the size changes over time.

Table S1 should be supplemented by KN99α

Response: Thank you for the comment. KN99 was already included in the original table with its mating type α indicated in the adjacent column; however, to improve clarity, we have now added the mating type directly next to the strain name as well.

References:

1. Fernandes KE, Fraser JA, Carter DA. Lineages Derived from Cryptococcus neoformans Type Strain H99 Support a Link between the Capacity to Be Pleomorphic and Virulence. Bahn Y-S, editor. mBio. 2022;13: e00283-22. doi:10.1128/mbio.00283-22

---

## [Editor Report · Decision Letter 2]

4 Dec 2025

Antifungal potential of Rhinacanthus nasutus extracts against the pathogenic fungus Cryptococcus neoformans

PONE-D-25-31554R2

Dear Dr. Chanarat,

We’re pleased to inform you that your manuscript has been judged scientifically suitable for publication and will be formally accepted for publication once it meets all outstanding technical requirements.

Kind regards,

Rajendra Upadhya

Academic Editor

PLOS ONE
---

## [Editor Report · Acceptance letter]

PONE-D-25-31554R2

PLOS One

Dear Dr. Chanarat,

I'm pleased to inform you that your manuscript has been deemed suitable for publication in PLOS One. Congratulations! Your manuscript is now being handed over to our production team.

Kind regards,

on behalf of

Dr. Rajendra Upadhya

Academic Editor

PLOS One